# Speculation of Sphingolipids in Capsanthin by Ultra-Performance Liquid Chromatography Coupled with Electrospray Ionization-Quadrupole—Time-of-Flight Mass Spectrometry

**DOI:** 10.3390/molecules28031010

**Published:** 2023-01-19

**Authors:** Mei-Li Xu, Lijun Qi, Xingfu Cai, Tong Cao, Rumeng Tang, Kaihang Cao, Yunhe Lian

**Affiliations:** 1Chenguang Biological Technology Group Co., Ltd., Handan 057250, China; 2Chenguang Biotech Group Handan Co., Ltd., Handan 056000, China; 3Hebei Natural Pigment Technology Innovation Center, Handan 057250, China

**Keywords:** ceramides, glucosylceramides, capsanthin, UPLC-Q-TOF HRMS/HRMS

## Abstract

Sphingolipids are constituents of cellular membranes and play important roles in cells. As nutraceutical compounds in foods, sphingolipids have been proven to be critical for human health. Therefore, the sphingolipids content of capsanthin was established based on ultra-performance liquid chromatography coupled with electrospray ionization-quadrupole–time-of-flight mass spectrometry. A total number of 40 sphingolipids were successfully identified, including 20 Glucosylceramides and 20 Ceramides. The predominant GlcCers contain 4-hydroxy-8-sphingenine t18:1 (**8**) with different structures of α-OH fatty acids. For the Cers, the main long-chain bases are 4-hydroxy-8-sphingenine t18:1 (**8**) and 4-hydroxysphingenine (t18:0) with different structures of α-OH or α, β-di (OH) fatty acids.

## 1. Introduction

Sphingolipids are a class of lipids existing in all living organisms as the important structural components of membranes. Plenty of research shows that sphingolipids play key roles in regulating the growth, differentiation, migration, and apoptosis of cells and the immune system in many mammalian cells [1,2,3,4,5,6,7]. Ceramides are a group of sphingolipids composed of sphingosine and a fatty acid via an amide linkage. Covalent linkage of the 1-hydroxyl group of the Cer to a carbohydrate moiety generates the highly complex glycosphingolipids, and the simplest neutral glycosphingolipids are cerebrosides with a glucose, glucosylceramide, or galactose, galactosylceramide sugar moiety.

Sphingolipids in mammalian organisms mainly contain trans-4-sphingenine d18:1 (4Z) or sphinganine d18:0 as a long-chain sphingoid base (LCB). They are also present in plants [8,9,10,11], and they contain mainly phytosphingosine t18:0 andt18:1 (**8**). Sphingolipids are present in high quantities in the human diet in the form of meat, milk, cheese, and soybeans [12]. In the human intestine, sphingolipids are digested into Cer by specific enzymes.

Valsecchi [13] characterized the Cers of Moro blood orange (*Citrus sinensis*), and the investigation of metabolic pathways and the composition analysis demonstrated that these Cers could be used as a nutraceutical. Capsanthin, as a natural pigment extracted from the fruit of capsicum, is widely used in the food, pharmaceutical, cosmetic, and feed industries. Sphingolipids as new effective components of the capsanthin, a systematic study was carried out, and 40 sphingolipids were identified, including 20 Cers and 20 GlcCers.

## 2. Results and Discussion

Recently, LC-MS/MS (liquid chromatography–tandem mass spectrometry) has been shown to be a useful tool for the structural characterization of sphingolipids [14,15,16,17,18,19,20,21]. In order to more efficiently separate and detect the complex sphingolipids in capsanthin, UPLC-Q/TOF-MS and MS/MS were coupled in this work.

Yamauchi [22] identified seven molecular species of GlcCers in red bell pepper (*Capsicum annuum* L.) by the method of HPLC purification and structural characterization with HPLC-MS and NMR. All these seven compounds appeared in our samples, and the spectrogram dates are consistent with the literature (e.g., compound **1**–**5**). Owing to Yamauchi’s work, sphingolipids can be identified by mass spectrometry during the current study, and a total number of 40 sphingolipids were successfully identified based on the molecular mass, including 20 glucosylceramides and 20 ceramides.

Figure 1 depicts the UPLC-MS chromatographic separation of the capsanthin sample. The signal generated by each ion transition uniquely identified molecular species by retention time, Ms1 and Ms2 spectra. Under our UPLC conditions, molecular species of sphingolipids have a good separation, indicating the reliable of the method to the capsanthin sample. Additionally, we were able to obtain definitive information on the sphingolipid’s structures from the MS analyses.

The protonated molecules of these sphingolipids were detected together with several product ions, such as the ceramide moiety, owing to the loss of the sugar moiety and the α-hydroxy fatty acyl moieties because of extensive in-source fragmentation during an APCI positive mode. Figure 2 shows the MS and MS/MS spectra of the GlcCer compound **13**, supposed to be the compound of GlcCer t18:1(8Z)/C16:0 8Z-N-2’-Hydroxytetracosanoyl-1-O-β-D-glucopy-ranosyl-4-hydroxy 8-sphingenine. The fragment ions can be classified into the following three groups:Fragments formed by loss of small neutrals (*m*/*z* = 826.6739, 682.6333, 664.6222, 646.6099, 628.6022);Fragments referring to the LCB (*m*/*z* = 298.2732, 280.2634, 262.2534);Fragments referring to the acyl component (*m*/*z* = 384.3823).

Figure 3 illustrates the supposed structures of the fragments. The interesting fragment *m*/*z* = 250.2516 (loss of 48 Da of 298.2732), occurring as a very wake peak, could be explained by the formation of a ring structure induced by the double bond in the LCB. The fragments at *m*/*z* 280.2634 (Figure 1) formed by the loss of H_2_O demonstrated octadec-8-ene-sphingosine as the characteristic recognition of the sphingolipids. The fragments at 280.2634 and 282.2793 provided structural information on the degree of unsaturation of LCB. Table 1 shows the “fingerprints” of the major GlcCers species.

The study by Whitaker [23] showed that both the 8-cis and 8-trans isomers of 4,8-sphingadienine and 8-sphingenine and cis-trans isomers of 4-hydroxy-8-sphingenine are present in the fruits, and glucose is the only sugar present in the bell pepper cerebrosides. In this study, the same precursor ion is always associated with two peaks having different UPLC retention times, which indicated the respective cis- and trans-sphingolipids.

The predominant GlcCers contain 4-hydroxy-8-sphingenine t18:1 (**8**) with different structures of α-OH fatty acids. For GlcCers, only the mono-hydroxylated FAs were detected for the GlcCers with alkyl chains ranging from 16 to 26 carbons. Figure 2 depicts GlcCers is about 40 percent of the total sphingolipids, calculating by area normalization method.

Figure 4 shows the MS and MS/MS spectrum of the Cer species compound **29**. The spectra of the Cer species show an analogous fragmentation, and the loss of small neutrals takes place in the same way as the GlcCer species. Table 2 and Table 3 show all precursor and product ions identified Cers of capsanthin. The amount of Cers is up to 50%, the main long-chain bases are 4-hydroxy-8-sphingenine t18:1 (**8**) and 4-hydroxysphingenine (t18:0). For the Cer compounds, with monohydroxylated and dihydroxylated FAs were detected, α-OH and α, β-di (OH), ranging from 16 to 26 carbons. Figure 2 depicts only a small number of fatty acids are α, β- di (OH), and very weak peaks of corresponding ions.

According to the role and the biological properties of the sphingolipids, Cer and GlcCer have potential applications as nutraceuticals and in the cosmetic industry. The nutritional value of capsanthin as a natural pigment extracted from the pepper was increased due to the existence of thtab.e sphingolipids. Therefore, the manufacturing technique for the production of sphingolipids from capsanthin or paprika is underway.

## 3. Materials and Methods

### 3.1. Chemicals and Materials

Acetonitrile and isopropyl alcohol of HPLC grade were purchased from Fisher Scientific (Waltham, MA, USA). Formic acid of HPLC grade was supplied by Anpu Scientific Instrument Co., Ltd. (Shanghai, China). All the water used was ultrapure water produced by a Master-S UVF ultrapure water system (Shanghai, China) and refreshed daily. The capsanthin samples were provided by Chenguang Biotech Group Co., Ltd. (Handan, China).

### 3.2. Pretreatment of Sphingolipids

The pepper particles (100 g) were extracted twice with 500 mL of a mixed solvent of n-hexane and acetone (9:1, *v*/*v*) at 40–45 °C for 1 h and then filtrated. The filtrate was liquid-liquid extracted three times with 600 mL 5% NaCl aq. at 40–45 °C, and thereafter purified the aqueous solution by extraction with 300 mL ethyl acetate. The aqueous solution was evaporated under reduced pressure to remove the water and obtain the sphingolipids as a brown powder.

### 3.3. UPLC Conditions

Reversed-phase HPLC analysis was performed on a Waters ACQUITY Ultra Performance LC system (UPLC, Waters Corp., Milford, MA, USA) using an ACQUITY UPLC BEH C18 analytical column (i.d. 2.1 mm × 150 mm, particle size 1.7 µm, pore size 130 Å, Waters) at 40 °C. The injection volume was 10 µL, and the flow rate was 0.2 mL/min. To achieve efficient separation of the ceramides, acetonitrile and isopropanol (1:1, *v*/*v*) containing 0.2 wt% of the formic acid was used as mobile phase A, and 0.2 wt% of formic acid aqueous solution was used as mobile phase B. The gradient elution program was as follows: first, with 60% A for 2 min, then with a linear gradient increasing to 84% A within 33 min, followed by a continuous linear gradient increasing to 100% A within 50 min and kept for 29 min, in the end, with a linear gradient reducing to 60% A within 1 min and held for 5 min. The total run time of the program was 120 min.

### 3.4. Mass Spectrometric Conditions

Mass spectrometry analysis was performed on a Waters Xevo G2-XS Q-TOF high-resolution mass spectrometer equipped with atmospheric pressure chemical ionization (APCI) ion sources. The instrumental parameters were set as follows: mass range scanned from 50 to 1200, high-purity nitrogen was used as nebulizer and drying gas. The flow rate of drying gas was 700 L h^−1^, and the source temperature was 110 °C. The samples were determined in positive-ion mode. The capillary voltage was set at 3.0 kV, and the sampling cone voltage was set at 40 V. MSE analysis was performed on a mass spectrometer set at 6 V. The scan times were 0.1 s and 1.0 s for UPLC-Q/TOF-MS and MS/MS, respectively.

## Figures and Tables

**Figure 1 molecules-28-01010-f001:**
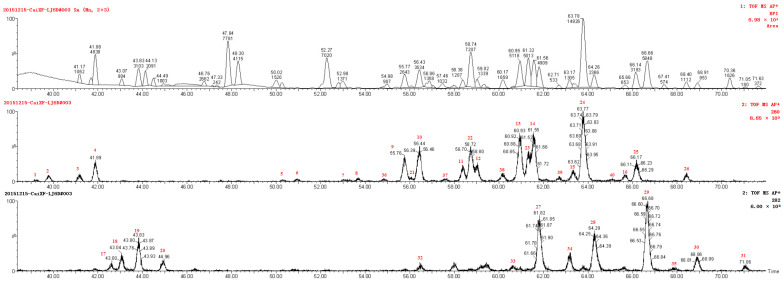
Chromatograms of UHPLC-Q-TOF/MS and compounds contain *m*/*z* = 280, 282 fragment ion peaks.

**Figure 2 molecules-28-01010-f002:**
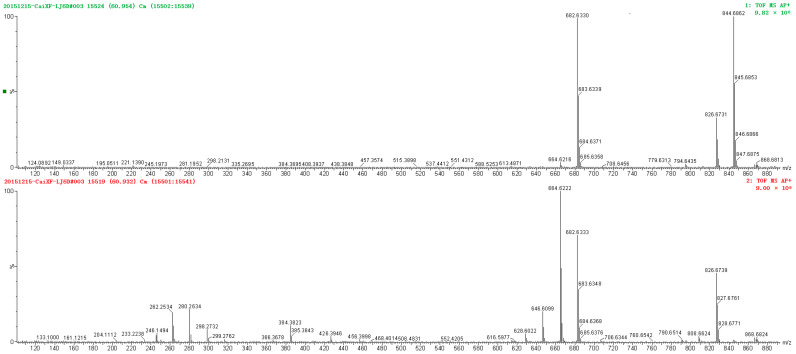
MS and MS/MS spectra of compound **13**.

**Figure 3 molecules-28-01010-f003:**
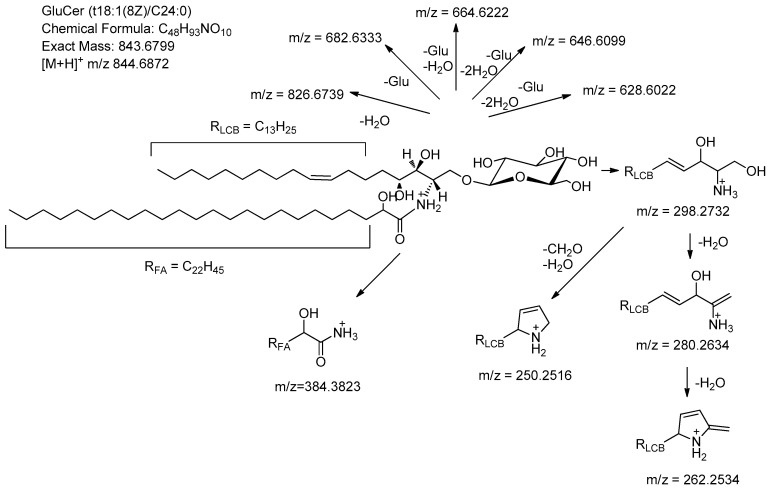
Proposed MS fragmentation scheme for GlcCer species compound **13**.

**Figure 4 molecules-28-01010-f004:**
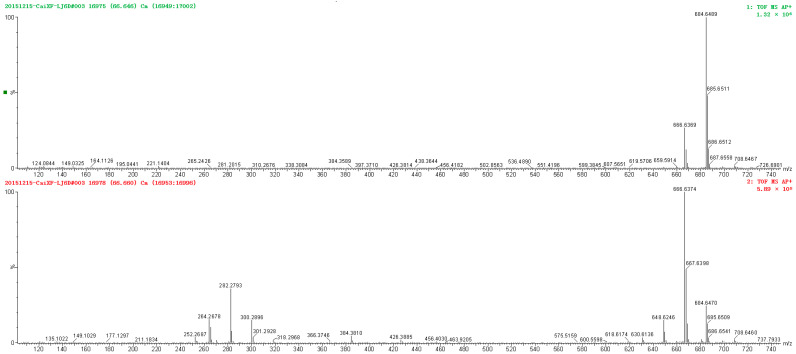
MS and MS/MS spectrum of Cer species compound **29**.

**Table 1 molecules-28-01010-t001:** Summary of precursor/product ion *m*/*z* values for glucosylceramides molecular species.

Peak NO. in Figure 4	Identity	Mol Formula	Retention Time (min)	Precursor Ion [M + H]^+^	Mass Deviation (ppm)	In-Source Fragments
**1** ^a^	GlcCer *t*18:1(8Z)/C16:0	C_40_H_77_NO_10_	39.240	732.5634	1.1	714.5470, 570.5038, 552.5008, 534.4952, 516.4750, 272.2622, 298.2640, 280.2631, 262.2523, 250.2230
**2** ^a^	GlcCer *t*18:1(8E)/C16:0	C_40_H_77_NO_10_	39.762	732.5577	−6.7	714.5466, 570.5014, 552.4971, 534.4833, 516.4709, 272.2516, 298.2710, 280.2632, 262.2488, 250.2545
**3** ^a^	GlcCer *d*18:2(4E,8Z)/C16:0	C_40_H_75_NO_9_	41.180	714.5504	−2.2	696.5394, 552.5018, 534.4865, 516.4765, 498.4711, 272.2555, 280.2615, 262.2527, 250.2527
**4** ^a^	GlcCer *d*18:2(4E,8E)/C16:0	C_40_H_75_NO_9_	41.887	714.5492	−3.9	696.5392, 552.4990, 534.4872, 516.4753, 498.4577, 272.2591, 280.2633, 262.2532, 250.2511
**5** ^a^	GlcCer *t*18:1(8Z)/C20:0	C_44_H_85_NO_10_	50.302	788.6201	−6.5	770.6164, 626.5701, 608.5527, 590.5428, 572.6311, 328.2327, 298.2710, 280.2494, 262.2310, 250.1502
**6**	GlcCer *t*18:1(8E)/C20:0	C_44_H_85_NO_10_	50.962	788.6236	−2.0	770.6119, 626.5787, 608.5562, 590.5421, 572.5375, 328.3295, 298.2754, 280.2599, 262.2542, 250.1935
**7**	GlcCer *t*18:1(8Z)/C21:0	C_45_H_87_NO_10_	53.003	802.6357	−6.4	784.6235, 640.5804, 622.5745, 604.5681, 586.5478, 342.3314, 298.2718, 280.2637, 262.2533, 250.2842
**8**	GlcCer *t*18:1(8E)/C21:0	C_45_H_87_NO_10_	53.706	802.6435	3.4	784.6340, 640.5870, 622.5740, 604.5605, 586.5483, 342.3280, 298.2476, 280.2622, 262.2554, 250.2282
**9** ^a^	GlcCer *t*18:1(8Z)/C22:0	C_46_H_89_NO_10_	55.743	816.6546	−2.3	798.6442, 654.6014, 636.5905, 618.5792, 600.5681, 356.3521, 298.2734, 280.2638, 262.2538, 250.2433
**10** ^a^	GlcCer *t*18:1(8E)/C22:0	C_46_H_89_NO_10_	56.414	816.6538	−3.3	798.6420, 654.6010, 636.5898, 618.5790, 600.5659, 356.3503, 298.2736, 280.2635, 262.2519, 250.2485
**11** ^a^	GlcCer *t*18:1(8Z)/C23:0	C_47_H_91_NO_10_	58.393	830.6686	−4.2	812.6581, 668.6169, 650.6047, 632.5930, 614.5784, 370.3656, 298.2737, 280.2629, 262.2523, 250.2486
**12** ^a^	GlcCer *t*18:1(8E)/C23:0	C_47_H_91_NO_10_	59.022	830.6694	−3.3	812.6564, 668.6173, 650.6068, 632.5977, 614.5822, 370.3662, 298.2727, 280.2620, 262.2531, 250.2401
**13** ^a^	GlcCer *t*18:1(8Z)/C24:0	C_48_H_93_NO_10_	60.932	844.6862	−1.9	826.6739, 682.6333, 664.6222, 646.6099, 628.6022, 384.3823, 298.2732, 280.2634, 262.2534, 250.2516
**14** ^a^	GlcCer *t*18:1(8E)/C24:0	C_48_H_93_NO_10_	61.558	844.6882	0.5	826.6769, 682.6355, 664.6232, 646.6137, 628.6021, 384.3827, 298.2742, 280.2648, 262.2551, 250.2624
**15** ^a^	GlcCer *t*18:1(8Z)/C25:0	C_49_H_95_NO_10_	63.338	858.7012	−2.6	840.6909, 696.6468, 678.6390, 660.6227, 642.6226, 398.3954, 298.2740, 280.2657, 262.2558, 250.2472
**16**	GlcCer *t*18:1(8Z)/C26:0	C_50_H_97_NO_10_	65.662	872.7170	−2.4	854.7078, 710.6650, 692.6543, 674.6449, 656.6318, 412.4139, 298.2817, 280.2620, 262.2531, 250.2545
**17**	GlcCer *d*18:1(4E)/C16:0	C_40_H_77_NO_9_	42.627	716.5623	−7.5	698.5519, 554.5049, 536.5037, 518.3867, 500.4689, 272.2453, 300.2813, 282.2692, 264.2675, 252.2696
**18**	GlcCer *d*18:1 (4Z)/C16:0	C_40_H_77_NO_9_	43.065	716.5671	−0.8	698.5558, 554.5139, 536.5027, 518.4926, 500.4810, 272.2585, 300.2824, 282.2780, 264.2684, 252.2589
**19** ^a^	GlcCer *d*18:1(8E)/C16:0	C_40_H_77_NO_9_	43.824	716.5661	−2.2	698.5563, 554.5127, 536.5024, 518.4901, 500.4734, 272.2595, 300.2853, 282.2785, 264.2689, 252.2610
**20** ^a^	GlcCer *d*18:1(8Z)/C16:0	C_40_H_77_NO_9_	44.941	716.5644	−4.6	698.5516, 554.5101, 536.5000, 518.4897, 500.4706, 272.2574, 300.2909, 282.2749, 264.2679, 252.2704

The prefixes “*d*” and “*t*” designate di- and trihydroxy sphingoid backbones, respectively. ^a^ Identified by Yamauchi [9].

**Table 2 molecules-28-01010-t002:** Summary of precursor/product ion *m*/*z* values for ceramides molecular species-α-OH.

Peak NO. in Figure 4	Identity	Mol Formula	Retention Time (min)	Precursor Ion [M + H]^+^	Mass Deviation (ppm)	In-Source Fragments
**21**	Cer *t*18:1(8E)/C21:0	C_39_H_77_NO_5_	56.061	640.5842	−5.9	622.5762, 604.5622, 586.5604, 342.3251, 298.2794, 280.2544, 262.2525, 250.2413
**22**	Cer *t*18:1(8E)/C22:0	C_40_H_79_NO_5_	58.731	654.6017	−2.9	636.5911, 618.5797, 600.5630, 356.3488, 298.2737, 280.2621, 262.2532, 250.2483
**23**	Cer *t*18:1(8E)/C23:0	C_41_H_81_NO_5_	61.332	668.6201	1.2	650.6070, 632.5964, 614.5864, 370.3588, 298.2745, 280.2654, 262.2546, 250.2499
**24**	Cer *t*18:1(8E)/C24:0	C_42_H_83_NO_5_	63.794	682.6343	−0.9	664.6221, 646.6118, 628.6007, 384.3823, 298.2739, 280.2634, 262.2530, 250.2474
**25**	Cer *t*18:1(8E)/C25:0	C_43_H_85_NO_5_	66.180	696.6473	−4.7	678.6379, 660.6263, 642.6121, 398.3926, 298.2733, 280.2634, 262.2524, 250.2519
**26**	Cer *t*18:1(8E)/C26:0	C_44_H_87_NO_5_	68.398	710.6653	1.3	692.6543, 674.6428, 656.6281, 412.4193, 298.2742, 280.2641, 262.2527, 250.2606
**27**	Cer *t*18:0/C22:0	C_40_H_81_NO_5_	61.823	656.6196	0.5	638.6069, 620.5929, 602.5778, 356.3482, 300.2935, 282.2811, 264.2675, 252.2646
**28**	Cer *t*18:0/C23:0	C_41_H_83_NO_5_	64.323	670.6332	−2.5	652.6213, 634.6117, 616.5888, 370.3677, 300.2863, 282.2784, 264.2646, 252.2683
**29**	Cer *t*18:0/C24:0	C_42_H_85_NO_5_	66.660	684.6489	−2.5	666.6374, 648.6246, 630.6136, 384.3810, 300.2896, 282.2793, 264.2678, 252.2687
**30**	Cer *t*18:0/C25:0	C_43_H_87_NO_5_	68.893	698.6639	−3.3	680.6516, 662.6420, 644.6274, 398.3915, 300.2901, 282.2774, 264.2668, 252.2619
**31**	Cer *t*18:0/C26:0	C_44_H_89_NO_5_	71.083	712.6754	−9.1	694.6643, 676.6531, 658.6328, 412.3976, 300.2846, 282.2782, 264.2603, 252.2670

**Table 3 molecules-28-01010-t003:** Summary of precursor/product ion *m*/*z* values for ceramides molecular species-α,β-(OH)2.

Peak No. in Figure 4	Identity	Mol Formula	Retention Time (min)	Precursor Ion [M + H]^+^	Mass Deviation (ppm)	In-Source Fragments
**32**	Cer *t*18:0/C22:0	C_40_H_81_NO_6_	57.956	672.6092	−7.4	654.5990, 636.5912, 618.5792, 600.5424, 372.3345, 300.2885, 282.2762, 264.2688, 252.2687
**33**	Cer *t*18:0/C23:0	C_41_H_83_NO_6_	60.725	686.6282	−2.5	668.6127, 650.6048, 632.5945, 614.5081, 386.3432, 300.2786, 282.2796, 264.2616, 252.2395
**34**	Cer *t*18:0/C24:0	C_42_H_85_NO_6_	63.192	700.6422	−4.7	682.6349, 664.6192, 646.6100, 628.5762, 400.3618, 300.2859, 282.2801, 264.2687, 252.2649
**35**	Cer *t*18:0/C26:0	C_44_H_89_NO_6_	67.868	728.6722	−6.3	710.6717, 692.6498, 674.5972, 656.6024, 428.3588, 300.3049, 282.2872, 264.2456, 252.2775
**36**	Cer *t*18:1/C22:0	C_40_H_79_NO_6_	54.852	670.5996	1.5	652.5925, 634.6226, 616.5663, 598.5677, 372.3235, 298.2316, 280.2595, 262.2215, 250.2215
**37**	Cer *t*18:1/C23:0	C_41_H_81_NO_6_	57.534	684.6113	−4.2	666.6025, 648.6174, 630.6168, 612.3077, 386.3209, 298.2623, 280.2640, 262.2560, 250.2477
**38**	Cer *t*18:1/C24:0	C_42_H_83_NO_6_	60.227	698.6283	−2.3	680.6160, 662.6033, 644.6029, 626.5577, 400.3712, 298.2746, 280.2657, 262.2518, 250.2541
**39**	Cer *t*18:1/C25:0	C_43_H_85_NO_6_	62.697	712.6404	−7.2	694.6312, 676.6130, 658.6057, 640.5693, 414.3831, 298.2746, 280.2602, 262.2528, 250.2536
**40**	Cer *t*18:1/C26:0	C_44_H_87_NO_6_	65.076	726.6617	0.7	708.6427, 690.6297, 672.6454, 654.5604, 428.4056, 298.2639, 280.2538, 262.2497, 250.2513

## Data Availability

The data presented in this study are available on request from thecorresponding authors.

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
