# Peer review of "Speculation of Sphingolipids in Capsanthin by Ultra-Performance Liquid Chromatography Coupled with Electrospray Ionization-Quadrupole—Time-of-Flight Mass Spectrometry"

_molecules, 2023, doi:10.3390/molecules28031010_

Round 1
Reviewer 1 Report
The manuscript presented a positive result in the identification of sphingolipids in capsanthin samples by UPLC-MS-MS techniques. The results maybe interested by the scientists working on capsanthin and related products. It can be considered to be accepted with some major revision. The detail comments are listed as below;
1. Is there any new sphingolipid identified from capsanthin sample?
2. From Figure 1 A, it is difficult to obtain result “GlcCers contain 4-hydroxy-8-sphingenine t18:1 (8)with different structures of α-OH fatty acids” quantitatively.
3. Many small peaks have been marked as known structure in Figure 1B, what is the peak in retention time about 60 min?
4. The advantage of this manuscript should be described. Methodology development in UPLC-MSn technique? Or new compounds?
Author contrubutions and Fundings should be rewritten.
Refernec 1 should be confirmed.
Author Response
Point 1: Is there any new sphingolipid identified from capsanthin sample?
Response 1: No new compounds were found in this study.
Point 2: From Figure 1 A, it is difficult to obtain result “GlcCers contain 4-hydroxy-8-sphingenine t18:1 (8)with different structures of α-OH fatty acids” quantitatively.
Response 2: I'm confused about this problem。This study has no quantitative work on “GlcCers contain 4-hydroxy-8-sphingenine t18:1 (8)”.
Point 3: Many small peaks have been marked as known structure in Figure 1B, what is the peak in retention time about 60 min?
Response 3: We have not resolved what compound is the peak in retention time about 60 min.
Point 4: The advantage of this manuscript should be described. Methodology development in UPLC-MSn technique? Or new compounds?
Response 4: As described in the introduction, sphingolipids play important roles in our life process, but so far, there are still few plant materials that can produce sphingolipids industrially. This study has proved the sphingolipids in pepper, and provided a method of industrialization。Therefore, the significance of this study is to provide more choices for the source of sphingolipids and provide technical reference for the comprehensive utilization of pepper.
Author contributions and Fundings have been rewritten.
References 1 have been confirmed.
Reviewer 2 Report
The paper describes the composition analysis of various sphingolipids in capsanthin using liquid chromatography/tandem mass spectrometry.
The method was performed with high resolution MS/MS and UHPLC.
And the speculation from product ions on MS/MS was investigated in detail.
However, I think that it has several revision points.
1) The word "identification" means that structural determination was completed. In this study, the authentic standards were not used. And so, "speculation" is better word in this study.
2) MS and MS/MS resolution were not described in the experiment section. It defines the need for the MS accuracy. The authors should reconsider that the need for the number of after the decimal point.
3) What is the strong point of using APCI mode?
4) The pretreatment method should be also described in the experiment section.
Author Response
Point 1: The word "identification" means that structural determination was completed. In this study, the authentic standards were not used. And so, "speculation" is better word in this study.
Response 1: Thank you for your suggestion, and the word of "identification" in tihs paper have changed to "speculation".
Point 2: MS and MS/MS resolution were not described in the experiment section. It defines the need for the MS accuracy. The authors should reconsider that the need for the number of after the decimal point.
Response 2: As described in the part 3 “Materials and Methods” , mass spectrometry analysis was performed on a Waters Xevo G2-XS Q-TOF High Resolution mass spectrometer. High Resolution mass spectrometer’s resolution >30000 (FWHM), and the number of after the decimal point is necessary.
Point 3: What is the strong point of using APCI mode?
Response 3: APCI is an atmospheric pressure atomization technology, which uses corona discharge to ionize the mobile phase, which can greatly increase the collision frequency of ions and sample molecules. Therefore, the atmospheric pressure chemical ionization sensitivity is three orders of magnitude higher than chemical ionization. It is one of the most commonly used atomization techniques in mass spectrometry. The substrate in this experiment has a good response under experimental conditions, so APCI atomization method is used in this experiment.
Point 4: The pretreatment method should be also described in the experiment section.
Response 4: The pretreatment method was added to the experimental section as follow.
Pretreatment of Sphingolipids
The pepper particles(100g)was extracted twice with 500 mL of mixed solvent of n-hexane and acetone (9:1, v/v) at 40-45 ℃ for 1h and then filtrated. The filtrate was liquid-liquid extraced three times with 600mL 5% NaCl aq. at 40-45℃, and thereafter purified the aqueous solution by extraction with 300mL ethyl acetate. The aqueous solution was evaporated under reduced pressure to remove the water and obtain the Sphingolipids as brown powder.
Round 2
Reviewer 1 Report
The manuscipt can be considered for publicaiton in current version.
Reviewer 2 Report
The paper describes the composition analysis of various sphingolipids in capsanthin using liquid chromatography/tandem mass spectrometry.
The paper has been improved the reviewer's comments.
I have no comments for more revision.
I recommend an acceptance.